# Associations between Autonomic and Endocrine Reactivity to Stress in Adolescence: Related to the Development of Anxiety?

**DOI:** 10.3390/healthcare11060869

**Published:** 2023-03-16

**Authors:** Jacqueline V. Stam, Victor L. Kallen, P. Michiel Westenberg

**Affiliations:** 1The Netherlands Organization for Applied Sciences (TNO), Department of Human Behavior & Training, Kampweg 55, 3769 DE Soesterberg, The Netherlands; 2Unit Developmental and Educational Psychology, Institute of Psychology, Leiden University, Wassenaarseweg 52, 2333 AK Leiden, The Netherlands

**Keywords:** mental health, stress, social anxiety, internalizing symptoms, public speaking task, adolescence, heart rate variability, cortisol, autonomic nerve system, HPA axis

## Abstract

Internalizing disorders in adolescence have been associated with disturbances in autonomic and endocrine functioning. Because the prefrontal cortex and the limbic system play a central role in regulating both the autonomic and the endocrine systems, their joint functioning is hypothesized to provide information about the potential development of internalizing symptoms throughout adolescence, notably in the preclinical stage. This hypothesis was tested in a sample of 198 adolescents from the general population. Heart rate variability (HRV) and skin conductance levels (SCLs) were measured before, during, and after a public speaking task. These autonomic parameters were associated with cortisol response to the task in the complete sample as well as in low- and high-anxiety adolescents separately. Self-reported social anxiety, low HRV, and high SCL recovery values were predictive of cortisol response. Importantly, in low-anxiety adolescents, only HRV during the task predicted the cortisol response, whereas, in their highly anxious peers, both HRV and SCL were strongly associated with this response. In the latter finding, age was a prominent factor. Additional analyses supported the idea that the interaction of autonomic and endocrine reactivity is subject to natural development. These findings provide evidence that adolescence might be a period of highly interactive emotional–neurobiological development, particularly with respect to the development of stress management skills.

## 1. Introduction

Disturbances in the functioning of both the autonomic nervous system (ANS) and the hypothalamic–pituitary–adrenal (HPA) axis have been associated with adverse emotional development in youth [1,2,3,4,5,6]. Only recently, interest in the interaction between the two systems has increased [7,8], although logically, the reactivity of these systems is at least reciprocal to some extent [9,10]. On the functional level, the locus coeruleus, activating the ANS, and the paraventricular nucleus of the hypothalamus are both innervated by corticotrophin and noradrenalin [11,12,13]. On the structural level, both systems are linked to the central autonomic network (CAN), including the prefrontal cortex and limbic structures (including, among other structures, the amygdala, the hypothalamus, and the hippocampus; see Figure 1) [14,15,16]. Considering the centrally coordinated reciprocity of the ANS and the HPA axis, the ability to adequately respond to a stressor might, consequently, not only be mirrored in typical singular responses within the autonomic or endocrine systems (e.g., increased heart rate, reduced heart rate variability, or increased cortisol production) but also in response patterns of the ANS and the HPA axis [9]. Following, though yet hypothetically, such response patterns might be indicative of the perceived severity of the encountered stressor and/or of the sensitivity of a given individual. Ambiguities reported in earlier studies that investigated the interaction between ANS and HPA axis responses [17,18,19] might very well be explained by these individual differences (see Figure 1).

During adolescence, brain development is characterized by functional changes in the limbic system and structural changes in the prefrontal cortex [20,21], going hand in hand with social reorientation and increased sensitivity to stress [22,23,24]. During this sensitive developmental period, stress systems (and their interactions) might consequently be particularly vulnerable to disturbances (i.e., excessive psychophysiological and/or neuroendocrinological responses reinforcing feelings of distress), potentially leading to adverse emotional development (reflected, for example, in feelings of anxiety or in avoidance behaviors) [23,25]. While maturating, it is even hypothesized that the HPA axis recalibrates during adolescence in response to social stress [26], making gradually developing deficiencies in HPA axis functioning a prominent risk factor for the development of, for example, anxiety disorders over time [27,28]. In short, adolescence is a vulnerable period for the development of stress regulation with potentially lifelong consequences for health and well-being [29,30]. In the current study, the development in adolescence of the relationship between autonomic and endocrine reactivity to a social stressor is studied, with particular interest in the potentially modulating roles of age and self-reported levels of social anxiety.

There is increasing evidence for the potentially interactive nature of the ANS and the HPA axis in response to stress in adults. This interaction seems to change before, during, and after exposure to stress. In stressful moments in a group of nurses, cortisol levels were positively associated with heart rate (as a measure of sympathetic activity of the ANS) and negatively associated with heart rate variability (HRV; as a measure of parasympathetic activity of the ANS [31,32,33]). Interestingly, these relations were absent in low stress periods [34]. In reaction to a standardized mental stress task, low baseline HRV in healthy men appeared to be associated with impaired poststress recovery of cortisol levels [35]. In contrast, higher baseline HRV has been associated with an increased cortisol response to the Trier Social Stress Test in young adults [36].

In younger children, previously reported associations between peripheral stress responses are generally nonsignificant. In toddlers (3 to 5 years), fearful temperament has been associated with an increased cortisol response and marginally lowered HRV changes in response to social stress [3]. Importantly, only nonsignificant relationships were reported between ANS parameters and the cortisol response, suggesting independent response profiles for the ANS and the HPA axis, at least in toddlers. This finding is confirmed by other studies suggesting a typical pattern of specifically parasympathetic reactivity in relation to endocrine responses to stress in youth [37]. However, in this somewhat older age group, there is some evidence linking patterns of sympathetic reactivity and HPA axis responses with inadequate coping and internalizing symptoms, such as distress, avoidance behavior, and/or lack of self-confidence in children [1]. This hypothesis is supported by an earlier study reporting a significant correlation between skin conductance level (SCL, as a measure of sympathetic arousal of the ANS) and cortisol levels in children, which appeared to be associated with (parent-reported) chronic internalizing problems (i.e., depression, anxiety, fear, worry, and psychosomatic symptoms) [38]. Interestingly, when examining baseline levels of HRV and cortisol, children with the highest levels on both measures were found to have the lowest levels of depression and anxiety symptoms [18].

Taken together, the nature of the interaction between the two stress systems might differ between immediate and chronic or repeated stress. Under immediate stress, as elicited by a public speaking task, reciprocal support of the ANS and the HPA axis may help to effectively deal with the challenge. However, a prolonged, exaggerated correlation between autonomic and endocrine responses may be indicative of rigidity in central coordination, e.g., by limbic structures such as the amygdala and/or hypothalamus (see Figure 1). This is a condition that has previously been associated with heightened feelings of stress and/or anxiety [35,39,40], as is the case with children and adolescents with social anxiety. Consequently, adequate coping might be reflected in independent or only moderately related reactivity of the ANS and the HPA axis in response to immediate stressors [14,41,42,43].

The aforementioned studies lead to three assumptions: (1) The autonomic response to a standardized stressor (an increase in SCL and a decrease in HRV) might be associated with the endocrine response (an increase in cortisol). (2) This association might be stronger in adolescents with higher self-reported levels of social anxiety. (3) The association may become more distinct with age. To investigate this, we used the data from the Social Anxiety and Normal Development study (SAND) to investigate SCL, HRV, and cortisol reactivity to a public speaking task in a sample of adolescents. We predict that (1) cortisol reactivity is negatively related to HRV and positively related to SCL reactivity; (2) this relationship is stronger in adolescents reporting higher levels of social anxiety; and (3) this relationship alters with increasing age.

## 2. Materials and Methods

### 2.1. Participants and Procedure

For the present analyses, previously collected data were used from the Social Anxiety and Normal Development (SAND) program of the Faculty of Social Sciences of Leiden University [44]. Participants for this study were recruited through high schools based on a normal distribution of age, gender, and school level, excluding individuals suffering from severe mental or medical conditions. Approximately 82% of the participants lived with both of their (biological) parents, and 11% lived with either a single mother or their mother and a new partner. Forty-nine percent of participants’ mothers completed tertiary education, indicating a normative SES distribution within the study [45].

Data from the 201 adolescents who participated in a public speaking task were used for the present study. Informed consent was obtained from the participants and their parent(s), and monetary rewards were provided after the session. The entire procedure was approved by the medical ethical committee of the Leiden University Medical Center and carried out in accordance with the Declaration of Helsinki.

The procedure entailed two laboratory sessions at the faculty: a presession and a public speaking session. In the presession, the participants filled out a series of questionnaires, among which was the Dutch version of the Social Anxiety Scale for Adolescents (SAS-A) [46]. The public speaking session was scheduled exactly one week after the presession. This gave all participants the opportunity to prepare their speech in advance about different kinds of movies. Participants were tested individually in separate rooms, each supervised by a trained assistant. The sessions commenced at 2:15 pm for all participants in order to minimize circadian rhythm effects on physiological data.

The public speaking session consisted of five consecutive phases: baseline, during which the participants watched an ocean wildlife DVD (first 25 min seated, then 5 min standing); anticipation, during which detailed instructions about the upcoming speech were given by the assistant through the intercom (3 min); preparation, during which the participants were instructed to prepare and/or rehearse their speech in silence (5 min); speech, during which they spoke in front of a prerecorded audience (5 min); and recovery, again seated watching an ocean wildlife DVD (10 min, starting approximately 10 min after the end of their speech). Detailed information regarding the recruitment and test procedure can be found elsewhere [44].

### 2.2. Assessments

Social anxiety: The Dutch version of the SAS-A contains 18 self-descriptive statements and 4 filler items. Each item is rated on a 5-point Likert scale. For each participant, an overall sum score of the 18 self-descriptive statements was calculated. The internal consistency of this scale is reported to be satisfactory [46,47]. In the present sample, Cronbach’s alpha was 0.94 (excellent).

Physiological parameters: A Bio-Pac ambulatory measuring system (MP150: Biopac Systems Inc., Goleta, CA, USA) was used to continuously measure heart rate and SCL. Heart rate was monitored using a precordial lead. Signals were amplified 1000 times and high pass filtered (0.5 Hz). SCL was measured by means of two Ag/AgCl electrodes positioned on the middle phalanxes of the forefinger and the middle finger of the nondominant hand. To avoid initial disturbances or potential movement artifacts, minutes 2 to 4 were selected from the last 5 minutes from the seated baseline period (baseline), the standing baseline period (baseline standing), the preparation period (preparation), the speech period (speech), and from the last 5 minutes of the recovery period (recovery). Based on beat-to-beat variations in heart rate, rMSSD (root mean square of successive differences between heartbeats) is widely used as parameter of HRV and, as such, accepted as indirect estimate of parasympathetic activity [48,49,50]. Because the rMSSD data were not normally distributed, we used the natural logarithm of rMSSD in the statistical analyses.

Endocrine parameters: During the procedure, eight saliva samples of at least 0.5 mL were collected to assess cortisol concentrations. This was performed just after arrival (*t* = 0), just after baseline (*t* = 30 min), just after the speech (*t* = 45), 10 min after the speech (*t* = 55), and then in four consecutive samples at 5 min intervals (*t* = 60, 65, 70, 75). Saliva samples were collected by passively drooling saliva into a plastic tube using a short plastic straw. Immediately after the procedure, the saliva samples were stored at −20 °C, and maximum one week later, the samples were stored at −80 °C. The determination of saliva cortisol concentrations was performed with a competitive electrochemiluminescence immunoassay ECLIA using a Modular Analytics E170 immunoassay analyzer from Roche Diagnostics (Mannheim, Germany). The lower detection limit was 0.5 nmol/L, and the coefficient of variation in the measuring range (4–80 nmol/L) was less than 10%. After determination of cortisol concentrations, three outliers were excluded based on their deviation (>3 sd), indicating possible contamination (most likely by blood). The area under the curve with respect to the ground (cortisol_AUC_) was calculated as parameter of endocrine reactivity [51].

### 2.3. Statistical Analyses

The median SAS-A score (41) was determined and used as splitting point for the formation of low- and high-anxiety groups. Chi-square statistics revealed no differences between the high and low SAS-A groups in terms of gender and age. MANCOVAs, with age as covariate and gender as a between-subjects factor, were applied to investigate associations with rMSSD and SCL in all periods. An ANCOVA was used to investigate the potential influence of age and gender on cortisol_AUC_.

To test the physiological reactivity to the public speaking task, we used repeated measures ANCOVAs including rMSSD and SCL, with SAS-A score as a between-subjects factor and the rMSSD or SCL levels during the seated baseline period as covariate. Helmert contrasts were used to investigate sequential changes over the experimental procedure. This procedure was repeated for the low and high social anxiety groups separately. Following these procedures, a t-test was applied to compare baseline cortisol levels with the highest post-task cortisol concentration.

For SCL and rMSSD in the baseline standing, preparation, speech, and recovery, partial correlations were used to test our first hypothesis of a relationship between autonomic functioning and cortisol_AUC_. Seated baseline values of either rMSSD or SCL were included as a covariate to correct for initial values. This procedure was repeated for the low- and high-anxiety groups separately.

To identify the relevant ANS predictors for cortisol_AUC_, multilevel regression analyses were conducted: one for the complete sample (with SAS-A score, age, rMSSD, and SCL in all periods as independent variables and cortisol_AUC_ as dependent variable), and one for the low- and high-anxiety groups separately (excluding the SAS-A scores as independent variable). Finally, we used a moving age window, each subsample containing 50 participants, with each sample increasing in age by removing the 10 youngest individuals and adding the 10 consecutive older participants, to calculate the correlation between baseline HRV with the cortisol response with increasing age.

## 3. Results

### 3.1. Descriptives

The eventual sample consisted of 100 boys and 98 girls (age range: 12.6–17.3 years, M = 14.8, sd = 1.31). The Social Anxiety Scale for Adolescents (SAS-A) scores ranged from 18 to 83 (M = 40.6, sd = 13.7). This mean value is below the mean level of normative data [46]. In the present sample, no gender differences for age and SAS-A scores were found. Of 194 participants, sufficient data were available to calculate cortisol_AUC_ (M = 69.2, sd = 35.3, range: 15.4–185.8).

Age and gender were unrelated to rMSSD or SCL in all of the selected periods. An ANCOVA including age as a covariate and gender as a factor showed no differences in cortisol_AUC_ between boys and girls, although age appeared to be positively correlated with cortisol_AUC_ (*r* = 0.16).

### 3.2. Autonomic and Endocrine Reactivity

The first repeated measures ANCOVA (with a Greenhouse–Geisser correction and seated baseline values included as a covariate) showed a significant rMSSD response over the procedure: (F(2.9, 550.3) = 3.7). Helmert contrasts showed a significant decrease in rMSSD between the standing baseline and the preparation period, and a significant increase in rMSSD after the speech toward the recovery period, see Figure 2. No effect of SAS-A score was found.

Changes in SCL over the procedure were found as well (Greenhouse–Geisser corrected: F(2.2, 421.2) = 101.3), although these findings disappeared when we corrected for the baseline (seated) levels of SCL. Only a marginally significant increase in SCL from the preparation to the speech period remained (see Figure 3). This finding appeared to be explained by the participants in the high SAS-A group, as they showed a marginal increase in SCL from the preparation period to the speech period (F(1, 90) = 3.5, *p* = 0.06), followed by an again marginal decrease toward the recovery period (F(1, 90) = 3.1, *p* = 0.08).

Cortisol responses to the task were significant. The cortisol concentrations at the end of the pretask baseline were lower than the highest cortisol concentration measured during the 25 min after the speech (t_197_ = 7.2, average increase: 2.3 nmol/L; average percentage increase: 143% above baseline value; see Figure 4). Importantly, no differences were found in cortisol_AUC_ between low and high SAS-A groups.

The results of the partial correlations between rMSSD in every consecutive period and cortisol_AUC_ are presented in Table 1. It appeared that rMSSD values before and during the public speaking task were all negatively correlated with cortisol_AUC_. The rMSSD recovery value was not. However, when we investigated this further to verify our second hypothesis, it appeared that the negative correlations between rMSSD and cortisol_AUC_ were very strong in the high SAS-A group, whereas in the low SAS-A group, only the decrease in rMSSD during the speech was significantly correlated with cortisol_AUC_ (see Table 1). No correlations were found between SCL and cortisol_AUC_ during any period, regardless of the group.

### 3.3. Social Anxiety and the Relationship between Autonomic and Endocrine Reactivity

Following the previous analyses, we conducted three regression analyses to find the strongest physiological predictors of cortisol_AUC_: one for the complete sample and two for the low and high SAS-A groups separately (see Table 2). In the total group, a combination of rMSSD and SCL data for multiple periods, together with age and SAS-A score, appeared to be predictive of cortisol_AUC_. However, in adolescents reporting relatively low social anxiety scores, only rMSSD during the speech remained a reliable predictor of cortisol_AUC_. Meanwhile, in adolescents reporting relatively high social anxiety scores, autonomic dynamics (i.e., both SCL and rMSSD during multiple periods) combined (together with age) into quite a powerful predictive model of cortisol_AUC_.

### 3.4. Association: Changes with Age

To visualize the contribution of age in the conducted regression analyses, correlations between baseline HRV and cortisol_AUC_ were calculated for every consecutive age cohort (consisting of 50 participants each; see Figure 5). Finally, we checked potential differences between age cohorts on this correlation using a Steiger analysis. It showed a significant difference in the correlation of baseline rMSSD and cortisol_AUC_ between the youngest age cohort (average age: 13.1 years) and the age cohort with the highest correlation (average age: 15.3 years), thereby confirming a change in the association between baseline rMSSD and cortisol response with age.

## 4. Discussion

The present study set out to investigate the interaction of autonomic and endocrine responses to a standardized laboratory stressor in adolescents using an adapted version of the Trier Social Stress Task [44]. Our additional aim was to find out whether and to what extent social anxiety plays a role in this interaction and whether this interaction is subject to increasing age.

It appeared that the cortisol response was associated with HRV reactivity, SCL recovery values, age, and self-reported social anxiety. In the low social anxiety adolescents (scoring below the median on the Social Anxiety Scale for Adolescents), only HRV during the task remained a predictor of the following cortisol response, albeit not a very strong one (R^2^ = 0.14). Contrary to this finding, in the high social anxiety adolescents (scoring above the median on the SAS-A), both HRV and SCL reactivity predicted cortisol response fairly well (R^2^ = 0.35). These results indicate qualitative and quantitative differences between high and low social anxiety adolescents. Finally, as age seemed to be a factor of relevance, we investigated whether the association between baseline HRV and cortisol response was subject to development. This appeared to be the case: in participants younger than 14 years, this relationship was absent. The relationship became significant until approximately 16 years and lost its interdependence again after the age of 16 years. This last finding suggests that midadolescence (14–16 years) may possibly be a period of heightened developmental sensitivity, especially in the reciprocal neurobiological management of stress under socially challenging conditions.

We consider these findings to be in line with previously published neurobiological models, such as the central autonomic network (CAN) [14] and the neurovisceral integration model [41]. These models suggest that autonomic reactivity to stress (lowered vagal tone resulting in lower HRV and heightened sympathetic activity resulting in higher SCL) should be directly related to the parallel-induced cortisol response. This physiological coping mechanism might be enhanced in individuals suffering from anxiety, as they are suggested to be less flexible in their physiological coping. This last hypothesis is based on the assumption that anxious individuals may suffer from a more ‘rigid’ stress regulatory network, likely caused by deficits in coordination and control on a more central level (i.e., amygdala and prefrontal cortex).

Overall, the present findings strongly support the relationship between generally lowered HRV and a heightened endocrine stress response in anxious adolescents in reaction to a social stressor. These findings seem to be in accordance with earlier findings [18,37,38] but in contrast with earlier findings in young children [3]. Exactly the age of the participant group may be the key to these discrepancies. Moreover, in the current study, we found associations from age 14 onwards. The associations between SCL and cortisol in the high social anxiety group are in line with earlier research showing an association of SCL with cortisol response in children with internalizing symptoms [38].

When interpreting the present data, some restrictions should be taken into account. Firstly, by design, no adolescents with clinical anxiety levels were included in the present sample. This was a logical choice, as we were primarily interested in the interaction of autonomic and endocrine dynamics in response to stress in adolescents. Nevertheless, extending the present findings with data from a clinical (adolescent) cohort would certainly be of relevance.

Secondly, the present findings might trigger an investigation into the (potential) role of the development of internalizing problems on the described neurobiological growth. Although we might suspect that low- and high-anxiety adolescents may show differentiated neurobiological developmental trajectories, a longitudinal study is necessary to investigate this hypothesis. Consequently, based on the present data, no absolute conclusions can be formulated regarding the potentially modulating role of anxiety in neurobiological development through adolescence.

Finally, in the current study, the stress task and the questionnaire were quite specific, aiming at inducing social stress and measuring social anxiety. This seemed to be a logical choice, as social anxiety appears to be the most prominent presentation of anxiety during adolescence [52,53]. However, it might be that the reported findings are not exclusively associated with social anxiety but also with other types of anxiety or internalizing symptoms. Consequently, a broader range of anxiety scales and stress tasks might be considered when planning future studies.

The present findings show that, in adolescents, autonomic responses to a social stressor are associated with the endocrine response. In low-anxiety adolescents, only the loss of parasympathetic (vagal) influence during the task is predictive of subsequent cortisol response. Interestingly, in high-anxiety adolescents, sympathetic and parasympathetic responses combine into a reasonably predictive model of the endocrine response. Irrespective of self-reported anxiety, the relationship between lowered baseline HRV and cortisol response to a social stressor appears to be subject to considerable developmental influence. These findings are in line with previously published results, both in children and in adults, and support prominent neurobiological theories emphasizing the reciprocal nature of autonomic dynamics and endocrine responses, especially in relation to internalizing symptoms.

## Figures and Tables

**Figure 1 healthcare-11-00869-f001:**
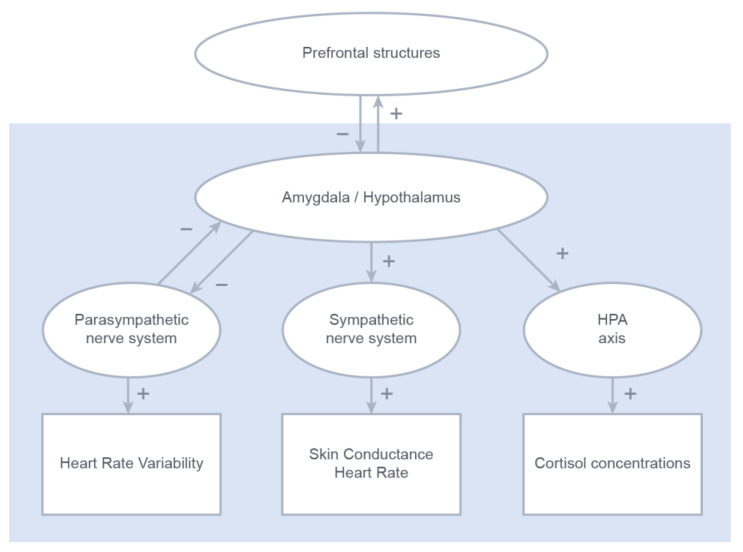
A schematic overview of the inhibitory and excitatory influences of multiple-layer neurological structures orchestrating the peripheral stress responses.

**Figure 2 healthcare-11-00869-f002:**
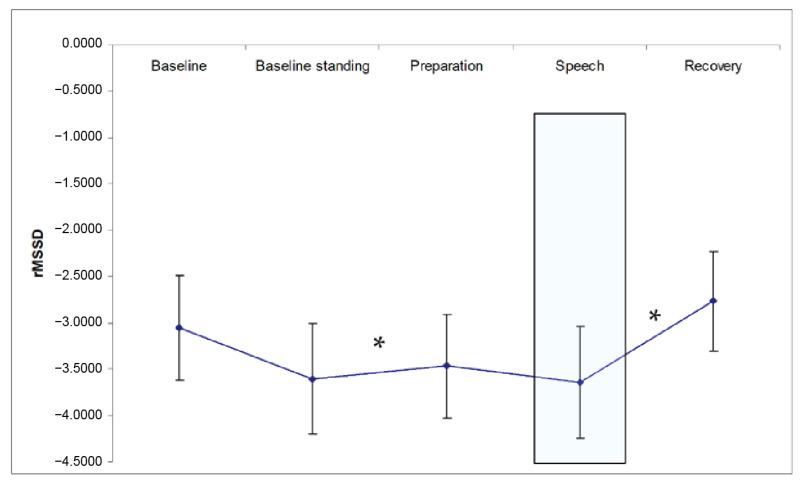
rMSSD (log-transformed) in response to the public speaking task. * *p* < 0.05 Helmert contrast (change).

**Figure 3 healthcare-11-00869-f003:**
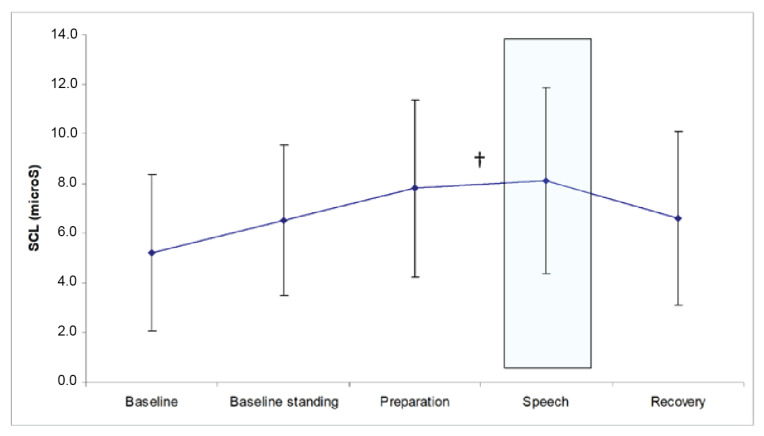
SCL in response to the public speaking task. ^†^
*p* < 0.10 Helmert contrast (change).

**Figure 4 healthcare-11-00869-f004:**
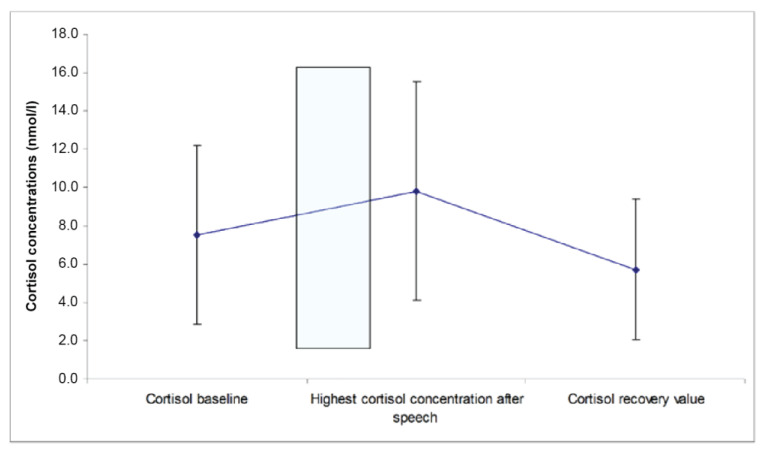
Cortisol response to the public speaking task.

**Figure 5 healthcare-11-00869-f005:**
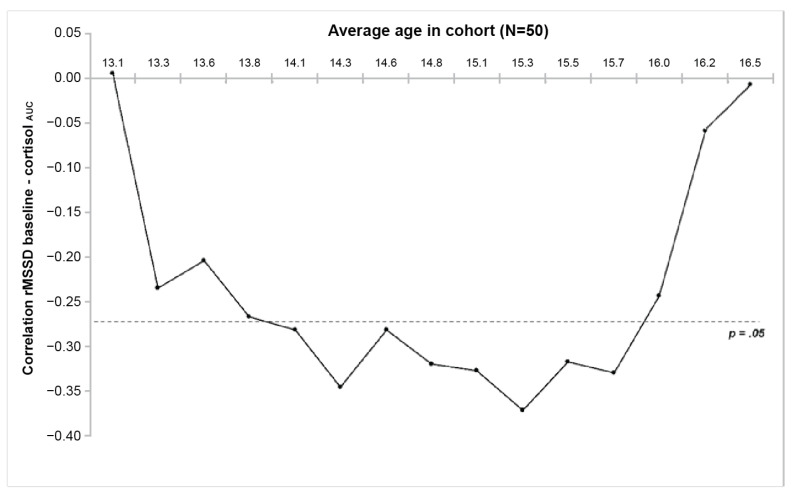
The correlation between baseline rMSSD and cortisol response to the public speaking task changes with age: significant correlations were found between 14 and 16 years of age.

**Table 1 healthcare-11-00869-t001:** Partial correlations (*r*, corrected for baseline values) of the log rMSSD and cortisol_AUC_.

	Total Sample *r*	Low Social Anxiety *r*	High Social Anxiety *r*
Baseline standing	−0.28 **	n.s.	−0.36 **
Preparation	−0.31 **	n.s.	−0.47 **
Speech	−0.33 **	−0.28 *	−0.39 **
Recovery	n.s.	n.s.	n.s.

* *p* < 0.05; ** *p* < 0.005.

**Table 2 healthcare-11-00869-t002:** Predictors of cortisol_AUC_ in the total sample and in the low- and high-anxiety groups.

Sample	Full Model Statistics	Included Independent Variable	β
Total sample	F (6, 183) = 9.4 ** Explained variance = 24%	Age	2.27 *
		SAS-A score	−1.91 ^†^
		rMSSD preparation	−1.71 ^†^
		rMSSD speech	−2.77 *
		SCL preparation	−2.58 *
		SCL recovery	2.58 *
Low social anxiety	F (1, 94) = 14.5 ** Explained variance = 14%	rMSSD speech	−3.80 **
High social anxiety	F (6, 88) = 7.3 ** Explained variance = 35%	Age	2.01 *
		rMSSD preparation	−3.05 **
		rMSSD speech	−1.93 ^†^
		rMSSD recovery	2.12 *
		SCL baseline seated	1.67 ^†^
		SCL preparation	−1.74 ^†^

† *p* < 0.10; * *p* < 0.05; ** *p* < 0.005.

## Data Availability

Data are available upon request.

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
