# Peer review of "Associations between Autonomic and Endocrine Reactivity to Stress in Adolescence: Related to the Development of Anxiety?"

_healthcare, 2023, doi:10.3390/healthcare11060869_

Round 1
Reviewer 1 Report
1. A well-designed and written manuscript that summarizes associations between autonomic and endocrine reactivity to stress in adolescence.
2. The introduction part is well written and summarized. Need to update with minor details.
3. Author can present his/her hypothesis with graphical abstract comprising of disturbances in autonomic and endocrine functioning during adolescence and role played by PFC (prefrontal cortex and limbic system.
4. Page 2…Line 47-48… Author stated stress systems might be vulnerable for disturbances, potentially leading to adverse emotional development… please elaborate what are those vulnerable disturbances causing such adverse effects during adolescence.
5. Elaborate effects of stress on HPA axis dysfunction.
6. Internalizing symptoms in children (Line 73) and internalizing problems (i.e., depression, anxiety, fear, worry, and psychosomatic 77 symptoms) (Line 77-78) …...Internalizing means are these kids resistant to such symptoms or they just coped well.
7. Line no 81-82…...The nature of the interaction between the two stress systems might 81differ between immediate and chronic or repeated stress… among these public speaking falls under which category.
8. Line 151-153…. Authors mentioned the determination of saliva cortisol concentrations and as mentioned 3 outliers were excluded. What was the criteria used to exclude the outliers? Was it done by mean±2SD? Or depending on lower detection limit?
9. Overall, the explanation of the proposed hypothesis of this work is very loud and clear.
With minor edits this manuscript will be ready for acceptance
Author Response
- A well-designed and written manuscript that summarizes associations between autonomic and endocrine reactivity tostress in adolescence.
Thank you very much for your appreciation. It was quite an interesting (and challenging) academic exercise to bring these fields together.
- The introduction part is well written and summarized. Need to update with minor details.
In the revised version of our manuscript we have strengthened the argumentation, thankfully using the input provided by the reviewers. Please find more details below.
- Author can present his/her hypothesis with graphical abstract comprising of disturbances in autonomic and endocrine functioning during adolescence and role played by PFC (prefrontal cortex and limbic system.
A very useful suggestion indeed, thank you. We have introduced an new figure 1 (line 45), outlining the inhibitory and excitatory influences of the relevant neurophysiological structures.
- Page 2…Line 47-48… Author stated stress systems might be vulnerable for disturbances, potentially leading to adverseemotional development… please elaborate what are those vulnerable disturbances causing such adverse effects during adolescence.
Thank you very much for providing us with the opportunity to elaborate a bit further on the ins-and-outs of (disturbed) HPA axis functioning, both in the context of neurocognitive development as well as related to (in)adequate stress management. More specific and concrete information has been added to the original text (where appropriate with some extra references). See lines 59 to 69 in the revised manuscript for the details provided.
- Elaborate effects of stress on HPA axis dysfunction.
See previous comment. More details have now been provided on stress responsiveness of the HPA axis and its relation with emotional functioning, as well as the either typically or subtlety disturbed development through adolescence (the latter already previously being associated with the potential development of psychopathology like anxiety disorders).
- Internalizing symptoms in children(Line 73) and internalizing problems (i.e., depression, anxiety, fear, worry, and psychosomatic 77 symptoms) (Line 77-78) …...Internalizing means are these kids resistant to such symptoms or they just coped well.
We have provided more details to better emphasize the distinction between internalizing symptoms (typical mild feelings of anxiety, avoidance behaviors, and alike) and internalizing problems, potentially developing into (clinical) syndromes (anxiety, depression, psychosomatic syndromes). See line 91 – 99.
- Line no 81-82…...The nature of the interaction between the two stress systems might 81differ between immediate and chronic or repeated stress… among these public speaking falls under which category.
This has been clarified (see line 101 – 102).
- Line 151-153…. Authors mentioned the determination of saliva cortisol concentrations and as mentioned 3 outliers were excluded. What was the criteria used to exclude the outliers? Was it done by mean±2SD? Or depending on lower detection limit?
A few samples provided remarkably high cortisol concentrations (for saliva samples), what suggested potential contamination with -most likely- blood. For this reason they have been excluded. What is now specified (line 187 – 189).
- Overall, the explanation of the proposed hypothesis of this work is very loud and clear. With minor edits this manuscript will be ready for acceptance
Thank you very much for your suggestions, which clearly helped us to strengthen our manuscript.
Reviewer 2 Report
The authors address a highly relevant topic: how do stress response systems interact in people that display differential emotional responses such as high and low anxiety and how does interaction evolve during development? They address these questions in a substantial number of individuals with a selection of well-established clinical outcomes that mirror stress reactivity of the two best-studied response systems, the HPA and the ANS. The manuscript is well written and argues conclusively. Of course, one could criticize that the data analyzed here were part of a larger study and that the groups were not predefined by e.g. cut-offs but generated by median split. It would also be interesting to know how the ANS/HPA responses interact with depressive symptoms and what the response looks like in adolescents with mental health disorders. However, statistical analysis is sound and I consider it important to focus and understand healthy development before tackling disease.
Minor changes required: add table with basic data and sociodemographic information on study population (e.g. BMI, educational background of parents etc.), add information, when the study was conducted (year, time of the year). Recalculate results controlling for confounders of anxiety and ANS/HPA function (e.g. BMI).
There is one open and speculative question you might want to contemplate: what, if changes in stress systems interaction are not a sign of adverse emotional development and a deficit but lead to diversity among populations that is required in a group to be best prepared for survival? E.g. the more anxious keep an alert ANS/HPA to early sense danger and warn the group, while others can follow up on survival tasks of the group without being distracted by emotions?
Author Response
- add table with basic data and sociodemographic information on study population (e.g. BMI, educational background of parents etc.),
Thank you very much for your time and effort to evaluate our study.
For the present data analyses, data was used from a larger longitudinal study on the development of social anxiety in adolescence (the SAND study, Leiden University). Relevant sociodemographic information of this study has been summarized in the revised version of our manuscript (Line 123 – 132). More details can be found (as indicated in the manuscript) in previously published, methodological, studies (Miers et al., 2013; Westenberg et al., 2009).
- add information, when the study was conducted (year, time of the year).
This has now been provided in more detail (see line 123 – 132).
- Recalculate results controlling for confounders of anxiety and ANS/HPA function (e.g. BMI).
In the context of the larger (SAND) study, all potentially relevant covariates have been investigated and tested (e.g. gender, age, menarche, menstrual cycle, SES, the use of contraceptives or nicotine, consumption patterns, BMI, etc.), which did not (significantly) influence the results or outcomes. Please see Miers et al., 2013; Westenberg et al., 2009 for more details.
Specifically the distribution of BMI within the present sample appeared to be relatively small (all but a few exceptions well within the general accepted healthy range, thus providing only little potential for covarying).
- There is one open and speculative question you might want to contemplate: what, if changes in stress systems interaction are not a sign of adverse emotional development and a deficit but lead to diversity among populations that is required in a group to be best prepared for survival? E.g. the more anxious keep an alert ANS/HPA to early sense danger and warn the group, while others can follow up on survival tasks of the group without being distracted by emotions?
Agreed, it would make very much sense that the presently found heterogeneity in responsiveness profiles contributes to the survivability of groups and communities. Another ‘tantalizing’ hypothesis obviously being associated with the present day type of typical challenges we encounter (behaving socially; need to be skilled in presenting in public; making deadlines and alike) and the original focus of our stress response neurophysiology.